# InfLLM: Training-Free Long-Context Extrapolation for LLMs with an Efficient Context Memory

**Chaojun Xiao**[1*]**, Pengle Zhang**[1*]**, Xu Han**[2,1,3†]**, Guangxuan Xiao**[4]**,**
**Yankai Lin**[5]**, Zhengyan Zhang**[1]**, Zhiyuan Liu**[1†]**, Maosong Sun**[1†]
[1]NLP Group, DCST, IAI, BNRIST, Tsinghua University
[2]Quan Cheng Laboratory [3]Shanghai Artificial Intelligence Laboratory
[4]Massachusetts Institution of Technology [5]Renmin University of China
`xiaocj20@mails.tsinghua.edu.cn`, `{hanxu2022,liuzy}@tsinghua.edu.cn`

## Abstract

Large language models (LLMs) have emerged as a cornerstone in real-world applications with lengthy streaming inputs (e.g., LLM-driven agents). However, existing LLMs, pre-trained on sequences with a restricted maximum length, cannot process longer sequences due to the out-of-domain and distraction issues. Common solutions often involve continual pre-training on longer sequences, which will introduce expensive computational overhead and uncontrollable change in model capabilities. In this paper, we unveil the intrinsic capacity of LLMs for understanding extremely long sequences without any fine-tuning. To this end, we introduce a training-free memory-based method, InfLLM. Specifically, InfLLM stores distant contexts into additional memory units and employs an efficient mechanism to lookup token-relevant units for attention computation. Thereby, InfLLM allows LLMs to efficiently process long sequences with a limited context window and well capture long-distance dependencies. Without any training, InfLLM enables LLMs that are pre-trained on sequences consisting of a few thousand tokens to achieve comparable performance with competitive baselines that continually train these LLMs on long sequences. Even when the sequence length is scaled to $1,024$K, InfLLM still effectively captures long-distance dependencies. Our code can be found at https://github.com/thunlp/InfLLM.

## 1 Introduction

Recently, large language models (LLMs) have achieved profound accomplishments in various tasks (Brown et al., 2020; Bommasani et al., 2021; Han et al., 2021; Touvron et al., 2023; Meta, 2024). Their ability to follow complex instructions shed light on the realization of artificial general intelligence (OpenAI, 2023; Ouyang et al., 2022). With the blooming of LLM-driven applications, such as agent construction (Park et al., 2023; Qian et al., 2023; Wang et al., 2024a) and embodied robotics (Driess et al., 2023; Liang et al., 2023), enhancing the capability of LLMs to process streaming long sequences become increasingly crucial. For instance, LLM-driven agents are required to process information continuously received from external environments based on all their historical memories, necessitating a robust capability for handling long streaming sequences.

Due to limitations caused by unseen lengthy inputs (Han et al., 2023) and distracting noisy contexts (Liu et al., 2023; Tworkowski et al., 2023), most LLMs, pre-trained on sequences consisting of only a few thousand tokens, cannot process longer sequences (Press et al., 2022; Zhao et al., 2023).

---

[*]Equal contribution.
[†]Corresponding authors.

38th Conference on Neural Information Processing Systems (NeurIPS 2024).

Common solutions usually involve continually training LLMs on longer sequences but further result in substantial costs and require large-scale high-quality long-sequence datasets (Xiong et al., 2023; Li et al., 2023). And the continual training process on longer sequences may weaken the performance of LLMs on short contexts (Ding et al., 2024). In view of this, improving the length generalizability of LLMs without further training receives extensive attention, trying to make LLMs trained on short sequences directly applicable to long sequences.

In this paper, we propose a training-free memory-based approach, named InfLLM, for streamingly processing extremely long sequences with limited computational costs. Specifically, InfLLM incorporate the sliding window attention (Xiao et al., 2023; Han et al., 2023) with an efficient context memory, where each token only attends to local contexts and relevant contexts from the memory. Considering the sparsity of attention score matrices, processing each token typically requires only a small portion of its contexts (Zhang et al., 2023b), and the remaining irrelevant contexts act as noise, leading to attention distraction issues (Tworkowski et al., 2023). We thus construct an external memory containing distant context information. Only relevant information within the memory is selected for each computation step, and other irrelevant noises are ignored. Owing to this, LLMs can understand whole long sequences using a finite-size window and avoid noisy contexts.

The vast amount of noisy context tokens in long sequences poses significant challenges to effective and efficient memory lookup. To address these challenges, we design a block-level context memory mechanism. Specifically, InfLLM organizes past key-value vectors into blocks, each containing a continuous token sequence. Within each block, the semantically most significant tokens that receive the highest attention scores are selected as the unit representation for subsequent relevance computation in memory lookup. This design offers two primary benefits: (1) Effective Lookup: The coherent semantics of each block can more effectively fulfill the requirements for relevant information retrieval compared to single tokens. The selection of unit representations minimizes the interference of unimportant tokens in relevance computation, enhancing the overall hit rate of memory lookup. (2) Efficient Lookup: The block-level memory unit eliminates the need for per-token relevance computation, significantly reducing computational costs. Moreover, block-level units ensure contiguous memory access, thus minimizing memory loading costs and enhancing computational efficiency. Furthermore, considering the infrequent usage of most units, InfLLM offloads all units on CPU memory and dynamically retains the frequently used units on GPU memory, significantly reducing GPU memory usage. Notably, the block-level memory mechanism in InfLLM does not involve any additional training, and can be directly applied to any LLMs.

To evaluate the effectiveness of InfLLM, we employ Mistral-7B-inst-v0.2 (Jiang et al., 2023) and Llama-3-8B-Instruct (Meta, 2024) as base models, which are pre-trained on the sequences containing no more than 32K and 8K tokens. We use two widely-used benchmarks, $\infty$-Bench (Zhang et al., 2023a) and Longbench (Bai et al., 2023), for evaluation. Especially, the average sequence length in $\infty$-Bench exceeds 100K tokens, which is challenging for most existing LLMs. Compared to typical methods that continually train LLMs on longer sequences, the experimental results demonstrate that InfLLM enables the LLMs pre-trained on the sequences containing a few thousand tokens to achieve comparable performance without any additional training. Moreover, we examine InfLLM on the sequences containing $1,024$K tokens, and InfLLM can still effectively capture long-distance dependencies, demonstrating the potential of InfLLM in scenarios involving long streaming inputs.

## 2   Related Work

Enabling LLMs to process long sequences has been extensively studied (Dong et al., 2023; Tay et al., 2023; Huang et al., 2023) and can generally be categorized into two main approaches: context length extrapolation and efficient context computation. The former aims to enable LLMs trained on short sequences to process much longer sequences. The latter focuses on enhancing the computational efficiency of attention layers, allowing efficient pre-training LLMs from scratch to process longer sequences. Although the focus of this paper is context length extrapolation, we also detailedly introduce efficient context computation. We also present the relevant works for memory-based models.

**Context Length Extrapolation.** Due to the high computational and memory requirements, the training of LLMs is often restricted to short sequences. Directly applying LLMs to long sequences will suffer from out-of-domain and distraction challenges caused by lengthy and noisy inputs (Han

et al., 2023; Tworkowski et al., 2023). Consequently, context length extrapolation has garnered attention as a method to improve the sequence length for LLMs without incurring additional training. The earliest approaches involve designing new relative positional encoding mechanisms during pre-training (Press et al., 2022; Sun et al., 2023). Subsequent studies mainly focus on the widely-used rotary position embedding (RoPE) (Su et al., 2021), and propose to achieve length extrapolation by downscaling or reusing the original position indices (Chen et al., 2023b; Peng et al., 2023; Chen et al., 2023a; Jin et al., 2024; An et al., 2024). These works can alleviate the out-of-domain issue from the unseen length, but can not alleviate the distraction challenge of noisy contexts. To address this, Xiao et al. (2023) and Han et al. (2023) employ the sliding window attention mechanism and directly discard all distant contexts to streamingly read extremely long sequences. However, as these models overlook information from distant tokens, they can not capture the long-distance dependencies for long-text understanding. In this paper, InfLLM utilizes the sliding window attention mechanism, and additionally constructs an efficient context memory to provide LLMs with relevant context information, enabling LLMs to effectively read and understand extremely long sequences.

**Efficient Context Computation.** The quadratic computational complexity of the attention layers is a primary factor limiting the lengthy sequence-processing capabilities of LLMs. Thus, numerous scholars have endeavored to design efficient attention mechanisms, including the utilization of sparse attention (Zaheer et al., 2020; Beltagy et al., 2020; Child et al., 2019; Ainslie et al., 2020; Zhao et al., 2019), approximating attention computations using kernel functions (Kitaev et al., 2020; Wang et al., 2020; Katharopoulos et al., 2020), and replacing the attention layer with linear-complexity state-space models (Gu et al., 2022; Gu & Dao, 2023). These approaches necessitate a modification in the model architecture, requiring retraining the models. Simultaneously, many researchers enhance the inference efficiency by evicting useless key-value vectors to reduce computation (Zhang et al., 2023b; Li et al., 2024; Ge et al., 2023). These methods can not extrapolate the context window of LLMs without further training due to out-of-domain issues caused by unseen positions. Recently, some researchers begin to explore the intrinsic sparse attention patterns of long-context LLMs and discard the redundant attention computation for acceleration (Jiang et al., 2024).

**Memory-based Models.** Memory networks have been studied for decades, which are proven effective in providing models with additional knowledge and information storage capabilities (Graves et al., 2014; Weston et al., 2015; Sukhbaatar et al., 2015; Miller et al., 2016). With the success of pre-trained models, memory layers have also been gradually applied in the training processes of recurrent transformer layers, enabling models to process long sequences recursively (Dai et al., 2019; Rae et al., 2020; Khandelwal et al., 2020; Wu et al., 2022; Bertsch et al., 2023; Munkhdalai et al., 2024). These works split sequences into segments, encoding each segment individually, and use memory to store context information from preceding segments. While these approaches are similar in concept to InfLLM, they involve modifications to the model architecture and requires further training the whole model. Besides, most existing memory-based methods focus on token-level memory units (Wu et al., 2022; Bertsch et al., 2023), which require a lot of time to build retrieval indexes for large-scale tokens in each input long sequence. Some methods also adopt block-level memory (Mohtashami & Jaggi, 2023; Tworkowski et al., 2023), these methods highlight the process of training effective block representations with long sequence data. In contrast, we aim to explore the inherent characteristics of LLMs, and propose a training-free memory module for long-text understanding.

# 3 Methodology

As shown in Figure 1, InfLLM builds a training-free context memory to efficiently provide highly-relevant contexts for each token, endowing the sliding window attention mechanism with the ability to capture long-distance dependencies.

## 3.1 Overall Framework

The main restrictions for improving the length generalizability of LLMs come from the out-of-domain and distraction issues caused by the lengthy and noisy contexts. To address these, following previous works (Xiao et al., 2023; Han et al., 2023), we adopt the sliding window attention mechanism, which only considers local tokens for each step. Additionally, we construct an extra context memory module to provide relevant context information to capture long-distance dependencies.

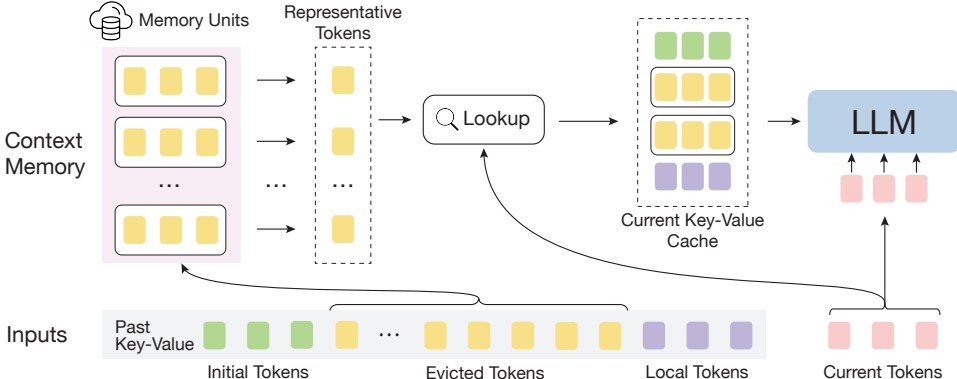

**Figure 1:** The illustration of InfLLM. Here, the current tokens refer to tokens that need to be encoded in the current computation step. The past key-value vectors can be divided into the initial tokens, evicted tokens, and local tokens, arranged the furthest to the nearest relative to the current tokens. For each computation step, the context window consists of the initial tokens, relevant memory units, and local tokens.

Specifically, we denote the long input sequence as $s = \{t_i\}_{i=1}^l$. Due to the limited GPU memory, instead of encoding the whole $s$ at once, we encode the input sequence $s$ chunk-by-chunk and generate the output token-by-token. For each computation step, the inputs consist of past key-value vectors $\mathbf{P} = \{(\mathbf{k}_j, \mathbf{v}_j)\}_{j=1}^{l_P}$ and current tokens $\mathbf{X} = \{\mathbf{t}_{i+l_P}\}_{i=1}^{l_X}$. For encoding steps, $l_X$ equals the chunk size, and for decoding steps, $l_X$ equals one.

According to the distances from current tokens, we can divide $\mathbf{P}$ into three groups: initial tokens, $\mathbf{I} = \mathbf{P}_{[1:l_I]}$, evicted tokens, $\mathbf{E} = \mathbf{P}_{[l_I+1:l_P-l_L]}$, and local tokens, $\mathbf{L} = \mathbf{P}_{[l_P-l_L+1:l_P]}$, arranged from the furthest to the nearest relative to the current tokens. Here, $l_P$, $l_I$, $l_L$ refer to the length of past key-value vectors, initial tokens, and the local window size. All evicted tokens, $\mathbf{E}$, are stored in the context memory, consisting of multiple memory units. For each step, InfLLM concatenates the initial tokens, relevant memories units from context memory, and local tokens to form the current key-value cache, $\mathbf{C} = \mathrm{Concat}(\mathbf{I}, f(\mathbf{X}, \mathbf{E}), \mathbf{L})$. $f(\cdot)$ refers to the lookup operation of context memory. The attention output is calculated as:

$$\mathbf{O} = \mathrm{Attn}\left[\mathbf{QX}, \mathrm{Concat}(\mathbf{C}_k, \mathbf{KX}), \mathrm{Concat}(\mathbf{C}_v, \mathbf{VX})\right].$$

Here, $\mathbf{Q}$, $\mathbf{K}$, and $\mathbf{V}$ are parameters in attention layers, $\mathbf{C}_k$ and $\mathbf{C}_v$ refer to the key and value vectors in $\mathbf{C}$. If $f(\cdot)$ always returns empty sets, InfLLM is degenerated into LM-Infinite (Han et al., 2023) and Streaming-LLM (Xiao et al., 2023), which directly discards distant contexts.

### 3.2 Context Memory

Previous findings indicate that the attention score matrices of LLMs are sparse, and we can generate the same outputs with only a small portion of key-value vectors preserved (Zhang et al., 2023b). Inspired by this, we design a context memory to efficiently look up relevant contexts from large-scale evicted tokens and ignore irrelevant ones to save computational costs. The most intuitive way is to construct a memory consisting of token-level memory units for every past key-value vectors, and every attention head separately, which would result in massive memory units, unacceptable computation, and non-contiguous memory access costs. Thus, considering the local semantic coherence of long sequences, we split the past key-value vectors into blocks, each serving as a memory unit, and conduct memory lookup at the block level to reduce the costs while preserving the performance.

In this subsection, we will introduce the details of the block-level memory units. Then we present the method to assign positional embeddings for selected relevant memory units and cache management for the context memory.

**Block-Level Memory Units.** Block-level memory units can save computation costs compared to token-level ones. It also poses new challenges for unit representations, which are supposed to contain the semantics of the entire unit for effective relevance score computation and be memory-efficient for context length scalability. Traditional methods usually involve training an additional encoder

to project a given unit into a low-dimension vector. Inspired by the token redundancy in hidden states (Goyal et al., 2020; Dai et al., 2020), we select several **representative tokens** from the entail blocks as the unit representation. For the $m$-th token, we define the representative score as:

$$r_m = \frac{1}{l_L} \sum_{j=1}^{l_L} \mathbf{q}_{m+j} \cdot \mathbf{k}_m,$$

where $\mathbf{q}_{m+j}$ is the query vector for $(m+j)$-th token and $\mathbf{k}_m$ is the key vector $m$-th token. Intuitively, $r_m$ represents the significance of the $m$-th token in its corresponding local window, indicating the extent of its influence on other tokens within the local window. The computation of representative scores requires no additional parameters.

Formally, given the evicted tokens, $\mathbf{E}$, we split it into several memory units, each containing $l_{bs}$ tokens. For each unit, the $r_k$ tokens with the highest representative scores are selected as representative tokens. Generally, $r_k$ is a small positive integer. Let us denote a memory unit as $\mathbf{B} = \{(\mathbf{k}_j^B, \mathbf{v}_j^B)\}_{j=1}^{l_{bs}}$, and the representative tokens of this unit as $R(\mathbf{B}) = \{(\mathbf{k}_{b_j}^B, \mathbf{v}_{b_j}^B)\}_{j=1}^{r_k}$.

For the **memory lookup** phrase, only $k_m$ units with the highest relevance scores are loaded for the current attention computation. We calculate the relevance score between $\mathbf{B}$ and current tokens $\mathbf{X}$ as:

$$\text{sim}(\mathbf{X}, \mathbf{B}) = \sum_{i=1}^{l_X} \sum_{j=1}^{r_k} \mathbf{q}_{i+l_P} \cdot \mathbf{k}_{b_j}^B.$$

Notably, the representative tokens selection is a training-free method to obtain the unit representations. Here, we can also train an additional encoder to generate more expressive unit representations, which we leave for future work.

**Positional Encoding.** Existing LLM training usually employs a finite number of positional encodings, which encounter out-of-domain distribution challenges when directly applied to longer sequence processing (Han et al., 2023). Besides, in InfLLM, the current key-value cache is composed of some discontinuous text blocks, and directly assigning continuous positional encodings to them would also lead to mismatch issues and confuse the model. Therefore, inspired by previous works (Raffel et al., 2020; Su, 2023), we assign all tokens beyond the local window size with the same positional encodings. Specifically, the distance between tokens in context memory units and current tokens is set as $l_L$.

**Cache Management.** To enable LLMs to process extremely long sequence streams while capturing the semantic relevance contained in the long contexts, we need to retain all memory units and look up them at each computation step. Considering the infrequent usage of most units, we employ an offloading mechanism, storing most memory units in CPU memory and only preserving the representative tokens and memory units needed in current steps in GPU memory. Additionally, given the semantic coherence of long sequences, where adjacent tokens often require similar memory units, we allocate a cache space in GPU memory, managed using a least recently used strategy. This approach allows for efficient encoding of extremely long sequences using limited GPU memory. From the observation, our offloading mechanism enables InfLLM to process sequences consisting of 100K tokens with only 26G VRAM. Besides, the miss rate of our GPU cache is quite low, which means the offloading mechanism does not introduce significant time overhead in memory loading while saving GPU memory usage. The details can be found in the Appendix.

Furthermore, for extremely long sequences, the representative tokens of each unit can also be offloaded to the CPU memory, constructing an efficient k-nearest-neighbor index, and thereby further reducing computational complexity.

## 4 Experiments

### 4.1 Settings

**Datasets.** We adopt representative tasks in a widely-used long document benchmark, $\infty$-Bench (Zhang et al., 2023a) for evaluation. We adopt the English datasets for evaluation as the base models are mainly pre-trained on English corpus. The datasets in $\infty$-Bench cover diverse tasks

**Table 1:** The results of InfLLM and baseline models on ∞-Bench. The 95% quantile for text lengths in ∞-Bench is 214K. The context window size for sliding window models refers to the local window size, and for InfLLM refers to "local window size + selected memory size".

| | Window | Streaming | R.PK | R.Num | R.KV | Choice | QA | Sum | Math.F | Avg. |
|---|---|---|---|---|---|---|---|---|---|---|
| Mistral-based Models (7B) | | | | | | | | | | |
| Mistral | 32K | ✗ | 28.8 | 28.8 | 14.8 | 44.5 | 12.9 | 25.9 | 20.6 | 25.2 |
| NTK | 128K | ✗ | 100.0 | 86.8 | 19.2 | 40.2 | 16.9 | 20.3 | 26.9 | 44.3 |
| SelfExtend | 128K | ✗ | 100.0 | 100.0 | 15.6 | 42.8 | 17.3 | 18.8 | 19.1 | 44.8 |
| Infinite | 32K | ✓ | 28.8 | 28.8 | 0.4 | 42.8 | 11.4 | 22.5 | 16.3 | 21.6 |
| Streaming | 32K | ✓ | 28.8 | 28.5 | 0.2 | 42.4 | 11.5 | 22.1 | 16.9 | 21.5 |
| H2O | 32K | ✓ | 8.6 | 4.8 | 2.6 | 48.0 | 15.6 | 24.4 | 26.9 | 18.7 |
| InfLLM | 16K | ✓ | 100.0 | 96.1 | 96.8 | 43.7 | 15.7 | 25.8 | 25.7 | 57.7 |
| Llama-3-based Models (8B) | | | | | | | | | | |
| Llama-3 | 8K | ✗ | 8.5 | 7.8 | 6.2 | 44.1 | 15.5 | 24.7 | 21.7 | 18.4 |
| NTK | 128K | ✗ | 0.0 | 0.0 | 0.0 | 0.0 | 0.4 | 6.4 | 2.6 | 1.3 |
| SelfExtend | 128K | ✗ | 100.0 | 100.0 | 0.2 | 19.7 | 8.6 | 14.7 | 22.6 | 38.0 |
| Infinite | 8K | ✓ | 6.8 | 7.6 | 0.2 | 41.5 | 14.6 | 20.8 | 20.6 | 16.0 |
| Streaming | 8K | ✓ | 8.5 | 8.3 | 0.4 | 40.6 | 14.3 | 20.4 | 21.4 | 16.3 |
| H2O | 8K | ✓ | 2.5 | 2.4 | 0.0 | 0.0 | 0.7 | 2.8 | 6.0 | 2.1 |
| InfLLM | 8K | ✓ | 100.0 | 99.0 | 5.0 | 43.7 | 19.5 | 24.3 | 23.7 | 45.0 |

including question answering, summarization, context retrieval, and mathematic computing. The average length for ∞-Bench is 145.1K. The 95% quantile for sequence lengths is 214K, which is far beyond the maximum length of the base models. Detailed statistics and task descriptions of these datasets are listed in the Appendix. Besides, we also conduct an evaluation on LongBench (Bai et al., 2023). The results for LongBench can be found in the Appendix.

**Baseline Models.** To verify the effectiveness of our proposed method, we compare InfLLM with the following competitive baseline models: (1) **Original** models: we present the performance of the original LLMs without context length extrapolation. (2) Position downscaling and resuing: NTK-aware scaled RoPE (**NTK**) (LocalLLaMA, 2023) designs a nonlinear interpolation method, which basically changes the rotation base of RoPE. **SelfExtend** reuse the position ids across neighboring tokens, which makes the extended relative positions in the scope of the training context window. (3) Sliding window: these methods apply the sliding window mechanism to discard distant contexts, including LM-Infinite (**Infinite**) (Han et al., 2023) and StreamingLLM (**Stream**) (Xiao et al., 2023). Therefore, for each attention computation step, the input length does not exceed the context window. (5) Key-value eviction: KV eviction methods aim to discard useless key-value vectors during long sequence processing and thus are usually used to reduce the computation complexity. We present the results of a widely-used key-value eviction method, **H2O** (Zhang et al., 2023b). The key-value eviction method cannot generalize to longer sequences due to the unseen position embeddings and is expected to achieve unsatisfactory performance.

Here, InfLLM and the models with the sliding window mechanism can be used to process extremely long streaming inputs. For NTK and SelfExtend, we extend the context window to 128K, which enables LLMs to process most instances in ∞-Bench.

### 4.2 Implementation Details

In this paper, we aim to enable LLMs trained with limited sequence length to read and understand extremely long sequences without further training. We adopt Mistral-7B-Instruct-v0.2 (Jiang et al., 2023) and Llama-3-8B-Instruct (Meta, 2024) as our base models. The maximum length of Mistral-7B-Instruct-v0.2 and Llama-3-8B-Instruct is 32K and 8K, respectively.

For our model, we set the encoding chunk size as 512, and the memory unit size for past key-value vectors, $l_{bs}$, as 128. The number of representative tokens, $r_k$, is set as 4. For both Mistral-based and Llama-3-based InfLLM, we set the local window size as 4K. For Mistral-based InfLLM, we load 96 relevant memory units for each step, and for Llama-3-based InfLLM, we load 32 relevant memory units. The number of initial tokens is set as 128 for LM-Infinite, StreamingLLM, and InfLLM to

**Table 2:** The comparison between InfLLM and models with continual pre-training, Llama-3-8B-Instruct-Gradient-1048k (Llama-1M). InfLLM can achieve comparable performance with Llama-1M with less computation consumption and memory usage.

| | Train-Free | R.PK | R.Num | R.KV | Choice | QA | Sum | Math.F | VRAM | Time |
|---|---|---|---|---|---|---|---|---|---|---|
| Llama-1M | ✗ | **100.0** | **99.8** | **23.2** | **51.5** | 13.6 | 18.5 | 18.3 | 76.6G | 40.4s |
| InfLLM | ✓ | **100.0** | 99.0 | 5.0 | 43.7 | **19.5** | **24.3** | **23.7** | **26.3**G | **26.7**s |
| Llama-1M+InfLLM | ✗ | 100.0 | 100.0 | 55.8 | 39.3 | 20.3 | 17.1 | 31.4 | 26.3G | 26.7s |

cover the system prompts and task descriptions. We adopt FlashAttention (Dao, 2023) to accelerate experiments for all baseline models. Please refer to the Appendix for more details.

### 4.3 Main Results

The results for Mistral-based models and Llama-3-based models are reported in Table 1. From the results, we can observe that: (1) Compared to models with the sliding window mechanism, which can also read extremely long sequences, our method demonstrates a significant performance improvement. This indicates that the context memory in InfLLM can accurately supplement LLMs with relevant contextual information, enabling efficient and effective understanding and reasoning on long sequences. (2) The position downscaling and resuing methods, NTK and SelfExtend, tend to compromise model performance while extending the sequence length to 128K. That is because these models cannot address the distraction issue caused by noisy contexts. In contrast, our model can consistently enhance performance for extremely long sequences. We successfully generalize Llama-3 from a 8K length to more than 16 times its length, achieving commendable performance on the ∞-Bench. (3) The position downscaling and resuing methods can increase the maximum sequence length of LLMs but also raise the computational and memory costs, limiting these methods' application. In contrast, InfLLM utilizes block-level memory and offloading mechanism, enabling efficient processing of long sequences within limited resources.

### 4.4 Comparing to Models with Continual Training

In this paper, we focus on expanding the context window of LLMs without additional training. In this section, we compare InfLLM with models that undergo continual training on long sequences in terms of both performance and efficiency. Specifically, we select Llama-3-8B-Instruct-Gradient-1048k (Llama-1M)[3], which have been further fine-tuned on long-text data and chat datasets, extending its context window to 1048K. Besides, we also employ InfLLM on the Llama-1M, where we set the local window as 4K and selected memory size as 4K. We present the results on ∞-Bench, the GPU memory usage, and time consumption in Table 2. From the results, we can observe that: (1) Compared to models that have undergone continual training on long sequences, InfLLM can achieve comparable or even superior results without any additional training. This suggests that LLMs inherently possess the capability to identify key information in long sequences and to understand and reason effectively. Notably, Llama-1M requires 512 GPUs for continual training, which is unaffordable for many researchers. In contrast, InfLLM does not require any training, which indicates the practicability of InfLLM. (2) In terms of efficiency, InfLLM achieves a 34% decrease in time consumption while using only 34% of the GPU memory compared to the full-attention models. Moreover, at longer sequence lengths of 256K tokens, the full-attention baseline fails due to out-of-memory errors, while InfLLM can efficiently process sequences up to 1024K tokens on a single GPU. (3) InfLLM can also be directly combined with the model with continual training and achieve comparable or even superior results with only 8K context window. It indicates that InfLLM can also serve as an efficient way to improve the inference speed.

### 4.5 Comparing to Retrieval-Augmented Generation

InfLLM leverages the intrinsic capacity of LLMs to construct a context memory for gathering token-relevant information, a concept similar to retrieval augmented generation (RAG) (Lewis et al., 2020; Nakano et al., 2021). However, compared to using RAG, where historical contexts are treated

---

[3]https://huggingface.co/gradientai/Llama-3-8B-Instruct-Gradient-1048k

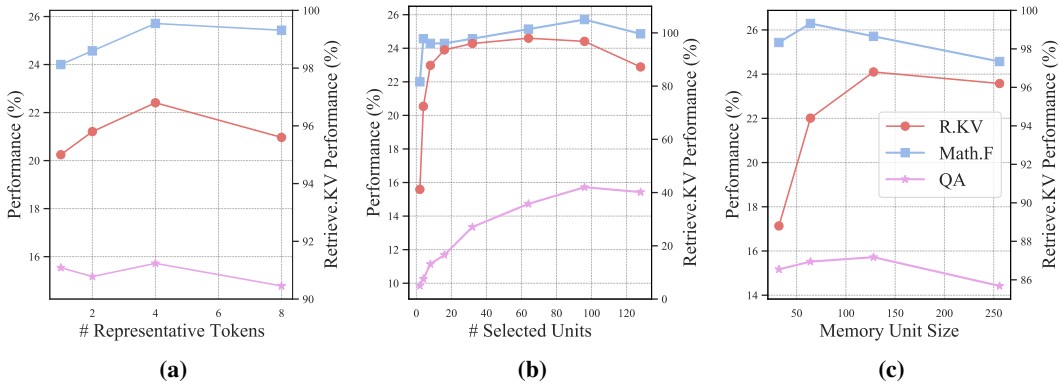

**Figure 2:** Extra studies about InfLLM. Here, (a), (b), and (c) investigate the impact of the context memory under different numbers of representative tokens, different numbers of selected units, and memory unit sizes, respectively.

as a searchable database for long-sequence understanding (Xu et al., 2023), InfLLM has several advantages: (1) Training-Free: RAG requires additional retrieval data to train a retrieval model, whereas InfLLM is training-free and applicable to any LLMs. Besides, RAG also necessitates fine-tuning LLMs to adapt to the inputs augmented by the retrieved knowledge. (2) Broader Applicability: RAG models are usually limited by the performance of their retrieval components. Besides, existing retrieval models will suffer from out-of-distribution issues, struggling to perform well on tasks outside their training distribution (Lin et al., 2023; Muennighoff et al., 2023). This limitation adversely affects the overall performance of the RAG system. In contrast, InfLLM has no specific requirements for tasks and can be feasibly used for long sequences.

To verify the generalization capabilities of InfLLM, we conduct experiments to comparing RAG and InfLLM on three context retrieval tasks. We utilize E5-mistral-7B-instruct (Wang et al., 2024b) as the retrieval model. The results are shown in Table 3. Our findings demonstrate that even without additional data or training, InfLLM can consistently outperform RAG models, underscoring its superior generalization capabilities. The dependency on an external retrieval model makes RAG less flexible in handling diverse tasks.

**Table 3:** The comparison between InfLLM and RAG.

| Task | R.PK | R.Num | R.KV |
|---|---|---|---|
| RAG-E5 | 89.2 | 65.4 | 13.2 |
| InfLLM | **100.0** | **96.1** | **96.8** |

### 4.6 The Impact of Memory Settings

InfLLM relies on the context memory to look up relevant information. We further explore the impact of core components in the context memory, specifically the representative tokens and memory units. The results are shown in Figure 2.

**Different Number of Representative Tokens.** InfLLM splits key-value vectors into memory units and selects several representative tokens from the unit to serve as the unit representations. Consequently, the ability of these representative tokens to semantically represent the entire unit directly impacts the model's performance. We conduct experiments with the number of representative tokens as $\{1, 2, 4, 8\}$. The results are shown in Figure 2a. It is observed that as the number of representative tokens increases, there is a trend of improvement in the model performance, which indicates that more representative tokens tend to better represent the semantic content of the memory units. However, it is noted that when the number of representative tokens reaches 8, there is a slight performance decrease. This decline can be attributed to the inclusion of semantically irrelevant tokens as unit representations. More efficient and powerful unit representations will further enhance model performance for future work.

**Different Number of Selected Units.** The selected units are utilized to provide relevant context to LLMs. We conduct experiments with the number of units set as $\{2, 4, 8, 16, 32, 64, 96, 128\}$. From Figire 2b, we can observe that as the number of selected units increases from 1 to 32, the model performance significantly improves, which is attributed to that more units imply a greater recall rate

of relevant content. Larger unit quantity also leads to an increase in the required memory scheduling time and the computational time for attention. Therefore, further enhancing lookup accuracy remains a crucial direction for improving the efficiency of InfLLM.

**Different Memory Unit Size.** Each memory unit is supposed to be a coherent semantic unit. Excessively large unit sizes can hinder precise lookup, while a small size will increase the computational overhead of memory lookup. We evaluate InfLLM with the unit size as $\{32, 64, 128, 256\}$ and keep the total context length as 12K. The results are shown in Figure 2c. It can be observed that the optimal unit size varies for different tasks due to the varying characteristics of input sequences. For example, in Retrieve.KV, a key-value pair constitutes a semantic unit, while in Math.Find, a single number represents a semantic unit. Employing heuristic rules to segment context can easily lead to suboptimal performance. Therefore, exploring how to dynamically segment context is an important direction for future research.

## 4.7 Ablation Study

To further verify the effectiveness of dynamic memory lookup and unit representations, we conduct ablation studies in this section. The results are shown in Table 4.

**Context Memory Lookup.** InfLLM adopts dynamic context memory lookup for both input encoding and output decoding steps for comprehensive long-text understanding. We present the results of InfLLM with only lookup in output decoding (Decoding-Only) and without any memory lookup (w/o Lookup). It can be observed that a significant decline in model performance is associated with a reduction in the number of memory lookup iterations. This indicates that distant contextual information is crucial for both

Table 4: The results for ablation study.

| Task | R.KV | Math.F | QA |
|------|------|--------|-----|
| InfLLM | **96.8** | 25.7 | **15.7** |
| Decoding-Only | 85.2 | **26.3** | 12.0 |
| w/o Lookup | 0.4 | 16.3 | 11.4 |
| Mean Repr | 84.6 | 25.1 | 14.9 |

the long-input encoding and answer-generation phases. The model requires the integration of long-distance context to generate a coherent context memory for input understanding. LLM is supposed to collect useful information from massive past context information to generate the correct answers.

**Unit Representation.** We design a block-level memory for efficient context information lookup. We select several representative tokens as the unit representations for relevance computation. We present the results of InfLLM with another training-free representation method (Mean Repr), which computes the representation by averaging the key vectors in a memory unit. From the results, we can observe that InfLLM with average representations can also present competitive performance. It indicates that the original attention vectors in LLMs are effective for relevance score computation, and exploring more efficient unit representations is an important future direction.

## 4.8 Scaling to 1,024K Context

To assess the effectiveness of InfLLM on extremely long sequences, in this subsection, we scale the sequence length to 1024K to evaluate the capacity of InfLLM to capture contextual relevance in long sequences. Specifically, we adopt the Retrieve.PassKey task in $\infty$-Bench for evaluation. This task prompts LLMs to find a 5-digit sequence among lengthy and noisy contexts, which requires LLMs to locate relevant information among long sequences effectively. We automatically generate inputs with $\{32, 64, 128, 256, 512, 768, 1024\}$ thousand tokens and for each length, we generate 50 instances for evaluation. We adopt Mistral as the base model.

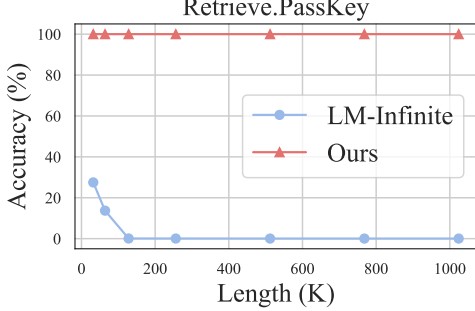

Figure 3: The results on sequences with different lengths.

The results are shown in Figure 3. From the results, we can observe that InfLLM can accurately locate the key information from length noises and achieve 100% accuracy even when the context length scales to 1024 thousand tokens. However, LM-Infinite can only attend to the tokens within the local window,

which leads to a rapid decline in its performance as the sequence length increases. It proves that InfLLM can accurately capture the long-distance dependencies for effective long-sequence reasoning.

## 5 Conclusion

In this paper, we propose a training-free method to improve the length generalizability of LLMs. Based on the sliding window attention mechanism, we construct an additional context memory module, which can help LLMs select relevant information from massive contexts to capture long-distance dependencies. The experiments on two widely-used long-text benchmarks show that InfLLM can effectively improve the ability of LLMs, which are trained on sequences with a few thousand tokens, to process extremely long sequences. In the future, we will explore efficient training of the context memory module to further enhance the model performance. Besides, combining the key-value cache compression methods with InfLLM can further reduce the computational and memory costs. We hope InfLLM can boost the development of streaming applications of LLMs.

## Acknowledgement

This work is supported by the National Key R&D Program of China (No.2022ZD0160501), Quan Cheng Laboratory (Grant No. QCLZD202301) and Institute Guo Qiang at Tsinghua University. Pengle Zhang is supported by Tsinghua University Initiative Scientific Research Program (Student Academic Research Advancement Program).

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

## Broader Impact

This paper presents work whose goal is to advance the field of long sequence processing for large language models. There are many potential societal consequences of our work, none of which we feel must be specifically highlighted here.

## Limitations

In this paper, we propose InfLLM, a method for extending the context window of LLMs without additional training. We verify the effectiveness of our model using a widely-used long-text evaluation benchmark ∞-Bench. However, our method still has the following limitations: (1) We store a large amount of past key-value (KV) cache in the CPU memory, which increases CPU memory usage. In the future, we can reduce CPU memory requirements by integrating techniques like KV cache quantization. (2) While InfLLM reduces the computational overhead for processing long texts in LLMs, there is still room for speed-up. In the future, we can further enhance the inference speed of InfLLM by integrating it with inference frameworks like llama.cpp[4] and vllm (Kwon et al., 2023).

## A    Cache Management Strategy

Due to the massive amount of memory units for extremely long sequences, we adopt an offloading mechanism to save GPU memory costs. Considering the infrequent usage of memory units, we offload most memory units to CPU memory and only preserve the frequently used memory units and current needed memory units in the GPU memory. To this end, we maintain a cache in GPU memory to effectively utilize GPU memory and reduce the communication between CPU and GPU. The size for our GPU cache is fixed, and therefore we design a least recently used (LRU) strategy for cache management. In this section, we will introduce the management strategy in detail.

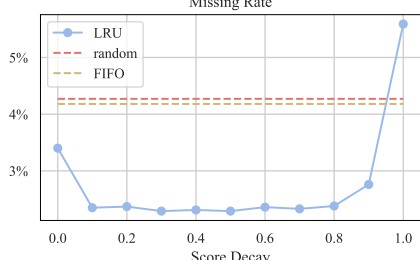

**Figure 4:** Missing rates of different cache management strategies.

**Loading Memory Units** For each computation step, we first compute the relevance scores for each memory unit to determine which units should be used. Then, for each needed memory unit, we first search it in our cache. If there is no hit, then we proceed with the transfer from CPU memory to GPU memory.

**Offloading Memory Units** After the attention computation, we need to offload redundant memory units to keep the GPU cache fixed. To this end, we apply an LRU strategy. Specifically, for each memory unit loaded into our GPU cache, we assign a frequency score $s_b$ for it, which will be used to determine whether this unit should be maintained in the GPU cache or offloaded to CPU memory to save GPU memory costs. The frequency scores are updated after the attention computation. Specifically, we update the score as follows:

$$s_b = s_b \cdot d + \sum_{j=1}^{l_X} \sum_{i=1}^{l_{bs}} \text{attention\_score}(\mathbf{q}_{j+l_P}, \mathbf{k}_i), \tag{1}$$

where $l_u$ represents the number of current tokens involved in this lookup, attention_score($\mathbf{q}, \mathbf{k}$) denotes the attention score between $\mathbf{Q}$ with respect to $\mathbf{k}$ (ranging from 0 to 1) obtained after performing the attention computation. $d$ is a hyper-parameter, representing the decay coefficient, used to incorporate the influence of previous lookups. After each attention computation, we sort all the memory units in our GPU cache according to their frequency scores $s_b$, and offload the units with the lowest scores back to the CPU memory.

To verify the effectiveness of our cache management strategy, we evaluate the cache missing rate of different cache management strategies on a sample of data from the GovReport dataset. Specifically, we compare our LRU strategy with (1) Random: randomly selecting units from the GPU cache to

---

[4]https://github.com/ggerganov/llama.cpp

**Table 5:** The results of InfLLM and baseline models on LongBench. The 95% quantile for text lengths in LongBench is 31K. The context window size for sliding window models refers to the local window size, and for InfLLM refers to "local window size + selected memory size".

| | Window | NQA | Qasper | MFQA | HQA | 2WikiMQA | Musique |
|---|---|---|---|---|---|---|---|
| Mistral-based Models (7B) | | | | | | | |
| Mistral | 32K | 22.06 | 29.16 | 47.65 | 37.53 | 21.96 | 19.03 |
| Infinite | 6K | 18.44 | 30.02 | 39.05 | 32.02 | 22.27 | 15.81 |
| Streaming | 6K | 17.92 | 30.05 | 39.09 | 32.18 | 21.83 | 14.71 |
| InfLLM | 6K | 22.12 | 29.33 | 47.42 | 36.56 | 22.31 | 17.68 |
| InfLLM | 12K | 23.03 | 29.52 | 47.62 | 39.53 | 23.61 | 18.92 |
| Llama-3-based Models (8B) | | | | | | | |
| Llama-3 | 8K | 19.85 | 42.36 | 41.03 | 47.38 | 39.20 | 22.96 |
| Infinite | 8K | 19.39 | 42.80 | 40.44 | 43.77 | 37.89 | 18.33 |
| Streaming | 8K | 20.05 | 42.46 | 39.54 | 43.69 | 37.89 | 19.68 |
| InfLLM | 8K | 22.64 | 43.70 | 49.03 | 49.04 | 35.61 | 26.06 |

| | Window | GovReport | QMSum | MultiNews | TREC | TQA | SAMSum |
|---|---|---|---|---|---|---|---|
| Mistral-based Models (7B) | | | | | | | |
| Mistral | 32K | 31.12 | 23.87 | 26.62 | 71.00 | 85.97 | 42.29 |
| Infinite | 6K | 29.74 | 21.92 | 26.65 | 70.00 | 85.22 | 41.60 |
| Streaming | 6K | 29.83 | 21.94 | 26.64 | 70.00 | 85.57 | 41.31 |
| InfLLM | 6K | 31.03 | 23.49 | 26.70 | 69.00 | 86.67 | 42.52 |
| InfLLM | 12K | 31.37 | 23.77 | 26.66 | 71.00 | 87.34 | 41.80 |
| Llama-3-based Models (8B) | | | | | | | |
| Llama-3 | 8K | 29.94 | 21.45 | 27.51 | 74.00 | 90.50 | 42.30 |
| Infinite | 8K | 29.25 | 21.41 | 27.62 | 74.00 | 90.08 | 41.72 |
| Streaming | 8K | 29.17 | 21.33 | 27.56 | 73.50 | 90.08 | 41.55 |
| InfLLM | 8K | 30.76 | 22.70 | 27.57 | 73.50 | 90.91 | 42.43 |

| | Window | PsgCount | PsgRetrieval | LCC | RepoBench-P | Avg. |
|---|---|---|---|---|---|---|
| Mistral-based Models (7B) | | | | | | |
| Mistral | 32K | 3.95 | 86.94 | 57.42 | 54.14 | 43.78 |
| Infinite | 6K | 2.08 | 42.80 | 57.12 | 53.43 | 39.07 |
| Streaming | 6K | 2.50 | 42.17 | 55.38 | 51.46 | 38.67 |
| InfLLM | 6K | 2.87 | 64.00 | 56.67 | 52.97 | 41.90 |
| InfLLM | 12K | 3.01 | 87.42 | 56.69 | 52.09 | 44.02 |
| Llama-3-based Models (8B) | | | | | | |
| Llama-3 | 8K | 8.50 | 62.50 | 60.83 | 49.14 | 44.73 |
| Infinite | 8K | 4.50 | 50.00 | 60.12 | 48.62 | 43.03 |
| Streaming | 8K | 5.00 | 49.00 | 60.35 | 48.95 | 42.99 |
| InfLLM | 8K | 7.17 | 84.00 | 59.88 | 46.48 | 46.95 |

offload. (2) First-in-first-out (FIFO): offload the unit that is first loaded in the GPU cache. The results are illustrated in Figure 4. It is observable that the LRU strategy we employed exhibits a lower missing rate, which ensures that the offloading mechanism does not introduce significant time overhead. In the experiments described in the main text, we chose a decay value of 0.1. Besides, to validate the effectiveness of our cache, we conducted an ablation study: running InfLLM without the GPU cache. The experimental results demonstrate that for encoding a 100K sequence, the addition of a GPU cache reduces our time costs from 21.5s to 18.8s.

# B  Positional Encoding

In InfLLM, we assign all tokens beyond the local window size with the same positional encoding. Therefore, for the current tokens, we do not explicitly provide positional information for the context. But we think that the unidirectional nature of a decoder-only model allows it to recognize the positional information of the context. For instance, assume a sequence contains three spans $S_A$, $S_B$, and $S_C$ in order. When encoding $S_C$, although $S_A$ and $S_B$ are assigned the same positional encoding, the unidirectional nature of the decoder-only model allows the key-value hidden states of $S_A$ and $S_B$

inherently embeds their relative positional information: $S_B$ can utilize information from $S_A$ during its encoding, while $S_A$ can only access information from preceding parts of the sequence.

To verify the model's capability to capture the relative positional information of the context, we adopt the Retrieve.Passkey task with multiple pass keys for evaluation. In this task, each sequence contains two pass keys, and the model is required to output these two pass keys in order. The data construction approach is consistent with that of $\infty$-Bench (Zhang et al., 2023a), where the positions of the two pass keys are randomly selected. We created 50 sequences, each 64K in length. The experimental results reveal that in this task, InfLLM can output the values of the two pass keys in the correct order 100% of the time. This indicates that, although our positional encoding disregards the relative positional information of the context, the model can still effectively understand the context in sequence.

## C   External Experiments

### C.1   Implementation Details

The context memory is constructed for all layers in LLMs. We set the size of our GPU cache as 32, which is twice the number of loaded units for each step. We set the frequency score decay coefficient as 0.1. We adopt the half-float precision for all experiments. We use NVIDIA A100 or A800 to conduct our experiments. For the experiment that scales to $1,024$K context, we set the encoding chunk size as 2048, and the number of representative tokens as 1 to speed up experiments.

### C.2   Performance on LongBench

We also employ LongBench Bai et al. (2023) as the benchmark to evaluate the effectiveness of InfLLM and baseline models. The evaluation results are shown in Table 5. The results indicate that: (1) InfLLM outperforms other models capable of processing streaming inputs across various diverse tasks. It proves that the context information provided by the context memory can efficiently enhance the model performance. (2) When applying Llama-3 as the base model, both StreamingLLM and LM-Infinite achieve only comparable or even worse performance than the original Llama-3. This indicates that while sliding window attention can effectively extend the context window size of LLMs, these models discard long-distance contextual information, thereby failing to achieve effective long-sequence understanding. (3) Mistral can handle text lengths up to 32K, covering most instances in LongBench. In contrast, InfLLM, with a window size of only 12K, achieves comparable or even superior performance on average. This further demonstrates InfLLM's ability to filter out noise in long contexts, leading to better long-sequence understanding.

### C.3   Experiments on Vicuna

In the previous sections, we demonstrated that InfLLM can extend the context windows of Llama-3 (with a maximum length of 8K) and Mistral (with a maximum length of 32K) to several hundred thousand tokens. To further validate the effectiveness of InfLLM, we apply it to the Vicuna Chiang et al. (2023), which has a maximum length of only 4K. The experimental results are shown in Table 6. The results show that we effectively extend Vicuna's context length to 128K, achieving significant performance improvements on the Retrieve.Passkey and Retrieve.Number tasks. However, InfLLM can not show performance gains on the Retrieve.KV and Math.Find tasks. This is because the hidden vectors contained in Vicuna have a limited ability to filter out noise in extremely long texts, making it difficult for context memory to effectively locate relevant information in the more complex contexts of the Retrieve.KV and Math.Find tasks. In the future, It deserves further exploration to design more powerful memory mechanism.

**Table 6:** The results of Vicuna-based models.

|        | R.PK  | R.Num | R.KV | Math.F |
|--------|-------|-------|------|--------|
| Vicuna | 5.08  | 4.41  | 1.40 | 11.71  |
| InfLLM | 99.15 | 81.69 | 0.60 | 11.14  |

**Table 7:** The combination of InfLLM and models with continual pre-training, Yi-9B-200K (Yi-200K).

| | Train-Free | R.PK | R.Num | R.KV | Choice | QA | Sum | Math.F |
|---|---|---|---|---|---|---|---|---|
| Yi-200K | ✗ | **100.0** | **98.3** | **54.5** | **63.3** | **13.0** | **5.9** | 23.4 |
| Yi-200K+InfLLM | ✗ | **100.0** | **98.3** | 47.8 | 45.4 | 8.2 | 4.7 | **33.1** |

### C.4 Combination of InfLLM with Yi-200K

We also present the results for the combination of InfLLM and Yi-9B-200K (Young et al., 2024) in Table 7. From the results, we can observe that InfLLM can also achieve comparable results with Yi-9B-200K.

