# OpenReview forum: "InfLLM: Training-Free Long-Context Extrapolation for LLMs with an Efficient Context Memory"
_NeurIPS.cc/2024/Conference — NeurIPS 2024 poster_

### Official Review · Reviewer_a1ZB · 2024-07-07

**Soundness:** 2
**Presentation:** 3
**Contribution:** 2
**Rating:** 5
**Confidence:** 5

**Summary:**

This paper proposes a key/value retrieval-based method for long-context processing in Transformers. To address the distraction issue in long-context processing, the authors introduce an efficient method that retrieves blocks of relevant keys and values for attention computation. Through extensive evaluations on long-context processing benchmarks, the authors demonstrate the efficiency and effectiveness of the proposed method compared to existing state-of-the-art baselines.

**Strengths:**

- The paper is easy to read.
- The proposed method is very simple yet surprisingly effective.
- The proposed method is training-free and can be applied to general Transformer models.
- The authors provide extensive experimental results, demonstrating the effectiveness of the proposed approach for long-context processing.

**Weaknesses:**

- The paper does not introduce new technical challenges, methods, benchmarks, or phenomena.
  - As I understand it, the proposed method adds a retrieval technique based on techniques from previous works (e.g., sink token, reassignment of positional encoding [1]).
  - Additionally, the block-wise attention mechanism is not new (e.g., landmark attention [2]), and some methods like the kNN-Transformer use advanced retrieval techniques such as Faiss [3], which can cluster and construct blocks with similar elements.

While I believe the paper is practically useful, the lack of novelty leads me to assign it a moderate score as a research paper.

[1] "Efficient Streaming Language Models with Attention Sinks", ICLR 2024
[2] "Landmark Attention: Random-Access Infinite Context Length for Transformers", NeurIPS 2023
[3] "Memorizing Transformers", ICLR 2022

**Questions:**

- Are the representative tokens different across layers? Can you observe any interesting patterns or insights regarding the representative tokens?
- I am not convinced why a memory unit size of around 100 achieves the best performance in Figure 2-(c). (i.e., Wouldn't the factual/contextual information be contained in fewer tokens than that?) In the figure, the authors fix the total context size, so a smaller memory unit size retrieves more memory units. What happens when we fix the number of retrieved units and vary the size of each unit?

**Limitations:**

Yes, in Appendix.

---

> ### Author Rebuttal · Authors · 2024-08-06
>
> We sincerely appreciate your thorough review of our paper. Here are some clarifications regarding the points raised:
> ### Q1: Novelty
> Please refer to Q1, Q2, Q3 in the Global Response. We implement an LLM with token-level memories, and it takes 2 hours to process a sequence with 128K tokens. The cost of time is unaffordable.
> We argue that building a memory for LLMs is the simplest and most effective method to address long-context challenges. Previous algorithms have struggled to achieve optimal efficiency and effectiveness. Long-text processing is a highly application-oriented field, and therefore, implementing a simple idea with high effectiveness is more conducive to real-world deployment. We believe our empirical methods can greatly benefit LLM deployment for long sequences.
> The simplicity of our approach, combined with its demonstrated effectiveness, makes it particularly valuable for practical applications. By focusing on a straightforward yet powerful memory mechanism, we have developed a solution that not only improves performance but also maintains feasibility for real-world implementation. This balance between simplicity and effectiveness is crucial in bridging the gap between theoretical advancements and practical deployment in the field of long-context language models.
>
> ### Q2: Representative Tokens
> Yes, different layers tend to choose different representative tokens. We conducted a case study on the selection of representative tokens, and from the results, we can broadly observe that in the top layers, the model generally selects words with stronger semantic information within spans. However, sometimes the model only selects common words, such as "the" or "a", as representations for spans. This indicates that there is still room for improvement in our unit representation approach.
>
> ### Q3: Memory Unit Size
> The segmentation of memory units is a trade-off between effectiveness and efficiency. When choosing a smaller memory unit size, it becomes easier to split a continuous statement into several semantically discontinuous spans, which can further interfere with retrieval effectiveness. Adopting a more dynamic span segmentation strategy is a future direction to improve InfLLM's performance. When we fix the number of retrieved units and vary the size of each unit, the results are as follows. As can be seen, when the unit size increases, the overall amount of relevant content retrieved also increases, leading to better performance.
>
> | Unit Size | R.KV  |
> | --------- | ----- |
> | 512       | 98.20 |
> | 256       | 96.00 |
> | 128       | 95.60 |
> | 64        | 93.20 |
>
> These findings suggest that larger unit sizes tend to capture longer contexts, resulting in improved performance. However, this also highlights the importance of balancing unit size with computational efficiency and the risk of including irrelevant information. Future work could explore adaptive unit sizing strategies to optimize this trade-off for different types of content and tasks.

---

> ### Author Response · Authors · 2024-08-09
> **Looking Forward to Further Discussion**
>
> Dear Reviewer,
>
> Thank you once more for your insightful feedback on our paper. The discussion period is now underway, and we welcome any additional questions regarding our responses. Reviewer 5j15 has acknowledged that our explanations addressed concerns about comparisons with memory-based methods and the novelty of InfLLM. We are keen to hear your thoughts on these issues as well. If you find that our responses have sufficiently addressed your concerns, we would appreciate a consideration for an improved score. Thank you again for your dedication and time.

---

> > ### Comment · Reviewer_a1ZB · 2024-08-12
> > **Thank you for rebuttal**
> >
> > In the rebuttal, the authors argue that "as the unit size increases, the overall amount of relevant content retrieved also increases, leading to better performance" (up to 512 tokens, which span approximately one page). However, it remains unclear why such an long context length is necessary to achieve the desired performance in their experiement settings. In L42 of the manuscript, the authors claim that "processing each token typically requires only a small portion of its context, and the remaining irrelevant context acts as noise, leading to attention distraction issues." I’m not convinced that their proposed method actually resolves this problem—it might simply be discontinous/OOD positional encoding issues. Given the weaknesses I've mentioned, I maintain the current score.

---

> > > ### Author Response · Authors · 2024-08-12
> > >
> > > Thank you for your feedback. As you noted, within certain limits, increasing the overall amount of relevant content retrieved does lead to better performance. However, as demonstrated in Figure 2(b) of the original paper, performance across all three tasks begins to decline when the selected units reach 128. This is due to the presence of increased noise within the context. Blindly increasing the content in the context does not always yield better performance.
> > >
> > > Regarding the memory lookup performance of InfLLM, we wish to emphasize, as detailed in Section 4.4, that InfLLM achieves comparable results to Llama3-1M with a 128K window using only an 8K window—just 6.25% of the window size. InfLLM has advanced the processing of long texts in a training-free setting with a limited context window effectively. However, we must acknowledge that InfLLM still requires several tens of retrieved blocks to achieve optimal results, indicating significant room for improvement in precisely locating the relevant memory units. Future exploration is needed, including dynamic block segmentation and more effective block representation.
> > >
> > > Nonetheless, these challenges do not negate the contributions of InfLLM. InfLLM provides an approach to handling long texts in a training-free manner: consistently utilizing a limited context window to select and retain crucial information from long sequences.
> > >
> > > I appreciate your attention to these points and look forward to your reconsideration based on these clarifications. We will add the discussion in the revision.

---

### Official Review · Reviewer_Srrj · 2024-07-10

**Soundness:** 2
**Presentation:** 3
**Contribution:** 2
**Rating:** 5
**Confidence:** 5

**Summary:**

This paper introduces InfLLM, which is a training-free memory-based method for long context extension. The key mechanism is to incorporate the sliding window attention with an efficient context memory. Each token only attends to local and relevant contexts from the memory. The paper conducts extensive experiments to demonstrate the effectiveness.

**Strengths:**

1. The paper is well written and easy to follow
2. The proposed method can have good/comparable performance with other baselines on Infinite-bench and longbench, while being much efficient

**Weaknesses:**

1. Although the paper conducts many experiments on LongBench and InfiniteBench, the two benchmarks do not reflect long capabilities. I suggest that the paper should also include ruler[1] and long context code understanding benchmarks[2].

2. The author makes comparisons with the training-based long context extension method and RAG in Sections 4.4 and 4.5 respectively, concluding that they can achieve comparable results but with better efficiency. However, the baseline models chosen by the author are not strong enough. Thus, the results and conclusion are not convincing. For example, there are many high-quality long context LLMs on 128k context window, please add them as baselines to evaluate InfLLM's performance within 128k context window.

3. The idea of using the initial token (e.g., streamingLLM) and selecting relevant tokens from past tokens for long context extension is incremental.

[1] https://arxiv.org/abs/2404.06654

[2] https://evalplus.github.io/repoqa.html

**Questions:**

See the weakness section

**Limitations:**

The main limitation of this paper is that the author claims that the training-free long context extension method can reach a 1M context window, and is comparable to training-based methods. This is counter-intuitive, and based on my own experience, long context extension requires fine-tuning for real application usage. However, there is no theoretical proof or analysis to support this, and the empirical experiments are not sufficient.

---

> ### Author Rebuttal · Authors · 2024-08-06
>
> We sincerely appreciate your thorough review of our paper. Here are some clarifications regarding the points raised:
> ### Q1: Evaluation on RULER
> Please refer to Q5 in the Global Response.
> ### Q2: Comparison to LLMs with 128K Context Window
> Please refer to Q4 in the Global Response. We compare InfLLM with the full-attention model based on Yi-9B-200K. From the results, we can observe that InfLLM can still achieve satisfactory performance, which further proves the effectiveness of InfLLM.
> ### Q3: Novelty
> Please refer to Q1, Q2, and Q3 in the Global Response.
> We argue that building a memory for LLMs is the simplest and most effective method to address long-context challenges. Previous algorithms have struggled to achieve optimal efficiency and effectiveness. Long-text processing is a highly application-oriented field, and therefore, implementing a simple idea with high effectiveness is more conducive to real-world deployment. We believe our empirical methods can greatly benefit LLM deployment for long sequences.
>
> The simplicity of our approach, combined with its demonstrated effectiveness, makes it particularly valuable for practical applications. By focusing on a straightforward yet powerful memory mechanism, we have developed a solution that not only improves performance but also maintains feasibility for real-world implementation. This balance between simplicity and effectiveness is crucial in bridging the gap between theoretical advancements and practical deployment in the field of long-context language models.

---

> > ### Author Response · Authors · 2024-08-09
> > **Looking Forward to Further Discussion**
> >
> > Dear Reviewer,
> >
> > Thank you for your ongoing engagement with our paper. As the discussion period unfolds, we welcome any new concerns regarding our responses. We were pleased to learn from Reviewer 5j15 that our explanations have adequately addressed key issues, including comparisons with well-trained long LLMs, broader benchmark evaluations, and the innovation presented in InfLLM. We eagerly anticipate your further feedback on these topics. If our replies have resolved your initial concerns, we kindly request an improvement of your initial scoring. We appreciate your commitment and time invested in reviewing our work.

---

> > ### Comment · Reviewer_Srrj · 2024-08-11
> > **Thanks for your rebuttal**
> >
> > Thanks for your detailed response! Although InfLLM performs better than streamingllm on the ruler, its performance on the ruler is still far behind other training-based long context approaches. I believe this might indicate that InfLLM is limited in practical long-context applications, so I will keep my original rating score.

---

### Official Review · Reviewer_5j15 · 2024-07-11

**Soundness:** 3
**Presentation:** 2
**Contribution:** 2
**Rating:** 7
**Confidence:** 4

**Summary:**

This paper focuses on the issue of extending context windows in LLMs by proposing a training-free block memory retrieval module. Specifically, aside from retaining initial tokens and local window tokens, other tokens are recalled using KNN at the block level. For efficient inference acceleration, LRU and chunk methods are employed to manage past key values in CPU or GPU memory. The proposed approach is tested on Mistral-32K and LLaMA-3-32K/1M models using benchmarks such as InfiniteBench and LongBench, with input tokens up to 214K. Results indicate that performance is generally maintained or improved across most tasks and models. However, there are notable performance drops in certain tasks when applying Long-context LLMs, such as the Choice task in Table 2.

**Strengths:**

- The research problem analysis in this paper is significant and highly applicable.
- The motivation behind the paper is sound. For each prompt request, relevant information is dynamic and sparse, which can be effectively handled through a KNN-based memory architecture.

**Weaknesses:**

1. The paper does not discuss the relationship and performance comparisons with other related memory-based methods[1-3], even though many require training.
2. While InfLLM performs well on most LLMs without extended context windows, Table 2 shows performance drops in some tasks for Long-context LLMs with InfLLM.
3. The datasets used in the experiments do not clearly reflect the changes in effective context window length when using InfLLM.
4. There is a lack of necessary ablation experiments to demonstrate the effectiveness of the block-level design.
5. The paper lacks results on end-to-end latency across different context windows.

- [1] Unlimiformer: Long-Range Transformers with Unlimited Length Input, NeurIPS 2023.
- [2] Focused Transformer: Contrastive Training for Context Scaling, NeurIPS 2023.
- [3] Leave No Context Behind: Efficient Infinite Context Transformers with Infini-attention.

**Questions:**

- **Q1**: Do you have any discussions or experimental results using other memory-based methods[1-3]?
- **Q2**: Do you have results using other well-trained Long-context LLMs, like Yi-200K[4]? Could you explain the significant performance drop in the Choice task in Table 2?
- **Q3**: Do you have any effective context windows benchmarks results, like Needle In A Haystack[5] or RULER[6]?
- **Q4**: Do you have the ablation results for the block-level design? Could you explain why performance drops with decoding only?
- **Q5**: Do you have the end-to-end latency results for different context windows?

[4] https://huggingface.co/01-ai/Yi-34B-200K

[5] https://github.com/gkamradt/LLMTest_NeedleInAHaystack

[6] https://github.com/hsiehjackson/RULER

**Limitations:**

Yes.

---

> ### Author Rebuttal · Authors · 2024-08-06
>
> We sincerely appreciate your thorough review of our paper. Here are some clarifications regarding the points raised:
>
> ### Q1: Comparison to Existing Memory-based Methods
> Please refer to Q1, Q2, and Q3 in the global response. Regarding the articles you mentioned, Unlimiformer employs token-level memory. From our experiments, token-level memory requires significant time for building the retrieval index. Even accelerated with Faiss, it still takes us 2 hours to process a 128K sequence. Focused Attention heavily relies on training unit representations, and the experimental results show that even with training, Focused Attention fails to achieve accurate retrieval on simple tasks like Passkey. Similarly, Infinite Attention uses linear attention mechanism to memorize distant tokens. However, this memory module suffers from catastrophic forgetting due to the limited memory state size, and experimental results demonstrate that this method cannot achieve general distant context memorization. Infinite Attention requires separate fine-tuning for different tasks, making it very impractical.
> In summary, InfLLM can achieve efficient and general context memory, enhancing LLMs' long-text processing capabilities without requiring training.
>
> ### Q2: Comparison to Well-trained Long LLM
> Please refer to Q4 in the Global Response. InfLLM can still achieve comparable performance with Yi-9B-200K, demonstrating InsLLM's effectiveness. Regarding the performance drop in the Choice task, we believe this is due to the nature of the task, which requires retrieving four separate context sections while encoding a short span (the four options). InfLLM's retrieval during the pre-filling stage is conducted on a block basis, making it challenging to retrieve information for four semantically distinct options. Therefore, further research into how to efficiently improve the granularity of InfLLM's retrieval during the pre-filling stage is a crucial area for future investigation.
>
> ### Q3: Evaluation on RULER
> Please refer to Q5 in the Global Response.
>
> ### Q4: Ablation Study for Block-level Design
> We implement an LLM with token-level memories, and it takes 2 hours to process a sequence with 128K tokens. The time cost is unaffordable, and thus we can not present the results for token-level memories. As for performance drops in decoding only settings, it indicates that prefilling stage is quite important for long sequence processing. With only sparse attention in the prefilling stage, LLMs cannot capture the semantic similarity between distant tokens. Similar phenomenon is also observed in MInference.
>
>
> ### Q5: Latency Across Different Context Windows
> In our current implementation, during the memory lookup process, we need to compute the dot product between the query and all unit representations. Therefore, in terms of complexity, our current implementation still has O(n^2) complexity. Recent research has shown that different attention heads exhibit significant sparsity (MInference). We can potentially combine InfLLM with these findings to further reduce inference complexity.
> In this paper, we emphasize InfLLM's context length extrapolation capability, which allows LLMs trained on shorter texts to better handle longer documents. Further reducing InfLLM's computational complexity is a direction for our future work.

---

> > ### Comment · Reviewer_5j15 · 2024-08-08
> > **Official Comment by Reviewer 5j15**
> >
> > Thank you to the authors for the detailed rebuttal! I truly appreciate your efforts in putting everything together within such a short period.
> >
> > My concerns have been addressed, and I have read the questions and answers from other reviewers as well. I have no further concerns. Overall, I believe InfLLM is a commendable work. It leverages the characteristics of natural language to design a block-wise, training-free retrieval method to extend long-context capabilities, resulting in improved latency and performance. The motivation is clear and the experiments are thorough.
> >
> > I have raised my score to 7.

---

> > > ### Author Response · Authors · 2024-08-08
> > >
> > > Thank you for your thorough and insightful feedback on our work. We greatly appreciate your kind words and the time you took to review our paper. Your positive comments and constructive suggestions have been invaluable in refining our research. We're glad that our efforts to build memory modules for enhanced long-context capabilities resonated with you.
> > >
> > > Thank you once again for your support and encouragement.

---

### Official Review · Reviewer_erR9 · 2024-07-13

**Soundness:** 3
**Presentation:** 1
**Contribution:** 1
**Rating:** 3
**Confidence:** 4

**Summary:**

This paper proposes breaking the context into blocks, and using a heuristic to compute representative tokens for the block. Then the attention score is first computed over these representative tokens, and the full attention is computed only over blocks corresponding to top-scoring representative tokens. The authors emphasize that the representative tokens are obtained without additional training. To improve memory usage, blocks are offloaded to CPU and only those selected for full attention are loaded into GPU. A cache is maintained to reduce offloading overhead.

**Strengths:**

The paper is well written. The results are reported on multiple tasks and models.

**Weaknesses:**

The paper does not cover related work well. As a result there is a lack of novelty when considering existing work. Many ideas such as breaking the context into blocks, using a representative to decide usefulness of a block, and offloading to CPU are used and discussed in earlier work such as [1]. The idea of using a heuristic for obtaining the representative of block is also proposed in [2]. Though a different rule (max-pooling) is used in that work. No comparison with either of these works is included in the paper. In particular, while the authors emphasize the training-free nature of the proposed method, it is not clear how much performance is lost in comparison with a training-based method.

Given the above discussion, the main contributions of this paper are the specific formula used to obtain the representative tokens, and the low-level offloading implementation together with cache. However, these contributions are not fully highlighted and evaluated in the current version. 1) While the importance of using LRU for cache eviction over other baseline strategies such as random eviction is briefly studied in terms of cache misses, the efficiency of offloading is not discussed. In particular, it is not clear whether actual speedup is observed when using offloading. Interestingly the rate for cache misses is very low and it would be interesting to see the traffic between CPU and GPU in this scenario.  2) The formula is not compared with alternative methods such as [2].

[1] Mohtashami A, Jaggi M. Landmark attention: Random-access infinite context length for transformers. arXiv preprint arXiv:2305.16300. 2023 May 25.
[2] Ren H, Dai H, Dai Z, Yang M, Leskovec J, Schuurmans D, Dai B. Combiner: Full attention transformer with sparse computation cost. Advances in Neural Information Processing Systems. 2021 Dec 6;34:22470-82.

**Questions:**

1. What is the overhead of offloading?

**Limitations:**

The authors mention avenues for future work. However, a disucssion around overheads and better/worse inference speed is missing.

---

> ### Author Rebuttal · Authors · 2024-08-06
>
> We sincerely appreciate your thorough review of our paper. Here are some clarifications regarding the points raised:
>
> ### Q1: Comparison to Existing Memory-based Methods
> Please refer to Q1, Q2, and Q3 in the global response. We are the first to construct a block-level training-free memory module for LLMs. Training-based methods require substantial computational resources and high-quality long-text pre-training and SFT data, which demand significant human effort. Moreover, in many real-world applications, such as ongoing conversations or LLM-driven agents, we need to process streaming input. In these scenarios, training-based methods are constrained by the maximum length used during their training process. Therefore, training-free context extrapolation is a crucial topic.
> Existing memory-based methods are mainly designed for pre-training language models from scratch. In contrast, InfLLM can be applied in a plug-and-play manner to all transformer-based methods, enhancing their ability to process long texts. Additionally, InfLLM is the first work to design a memory module specifically suited for training-free context extrapolation, implementing a comprehensive and efficient inference solution. Therefore, in this paper, we focus on the performance of memory-based methods in length extrapolation. In our revision, we will further discuss the differences between our work and existing memory-based methods.
>
> As for the comparison with training-based methods, please refer to Table 2 in the original paper and Table 1 in the global response. We compared InfLLM with full-attention models, which can be considered the upper bound of long-context LLMs. The experimental results show that InfLLM can perform comparably to these models on InfiniteBench, demonstrating the effectiveness of InfLLM.
>
> ### Q2: Efficiency in Offloading Mechanism
> As stated in our paper, due to the semantic continuity of long text sequences, the memory units required for adjacent text spans are often highly similar. Therefore, utilizing a GPU cache can effectively reduce the communication between GPU and CPU. Even we employ the random strategy to manage GPU cache, the missing rate is quite low. Therefore, our findings are that the offloading mechanism should be widely-used for long-context inference, which will not incur much additional time costs.
>
> To validate the effectiveness of our cache, we conducted an ablation study: running InfLLM without the GPU cache. The experimental results demonstrate that for encoding a 100K sequence, the addition of a GPU cache reduces our time costs from 21.5s to 18.8s.

---

> > ### Author Response · Authors · 2024-08-09
> > **Looking Forward to Further Discussion**
> >
> > Dear Reviewer,
> >
> > Thank you once again for your valuable comments on our paper. The discussion period has now started, and we welcome any further questions you may have regarding my responses. Reviewer 5j15 has indicated that our responses have resolved his/her concerns regarding the comparison with memory-based methods. We are also eager to hear your views on this matter. Should our responses have addressed your concerns, we would be grateful for an improved score. Thank you again for your time and efforts.

---

> > > ### Comment · Reviewer_erR9 · 2024-08-11
> > >
> > > Dear Authors,
> > >
> > > Thank you for your replies.
> > >
> > > Regarding comparison with existing methods: I understand that the cost of running some of these methods can be quite high. I also understand that proposing a method that is showing good performance can be generally useful. On the other hand, there is a growing number of proposed methods (each possibly targeting a slightly different setting, e.g. training-free) all of which report great accuracy on some tasks.  But without any comparison or ablation study it is impossible to identify the advantages of the proposed method.
> > >
> > > I made specific suggestions in my review such as using alternative formulas instead of the proposed formula, particularly those used in previous work such as [2] in my review (though [2] does not perform training-free lookup, the formula can still be used). The authors did not provide any reply on this suggestion. The idea is to understand whether that specific formula matters.
> > >
> > > Also thank you for pointing out that you are matching the full attention in Table 2 but this does not answer whether other methods would do so as well or not. In other words, it is not clear whether the task is not hard enough to distinguish the differences or the method is good.
> > >
> > > > Additionally, InfLLM is the first work to design a memory module specifically suited for training-free context extrapolation, implementing a comprehensive and efficient inference solution.
> > >
> > > I am not sure if this claim is justified. Similar not fully justified claims are often made throughout the paper as I mentioned in my original review. The idea of maintaining a cache and offloading to CPU is not novel and has been implemented before, for example in [3]. Various eviction policies have also been looked at for example in [4]. The combination with retrieving based on a representative has also been looked at before. That is what the memory module is doing.  Can you please specify what is the special novelty of the memory module that makes this the first work to design it? The low-level implementation is useful and I do not know any other work that has done it. But then this module needs to be fully benchmarked to show the impact on inference speed.
> > >
> > > As I mentioned in my original review, in my view the paper's novel contributions seem to be "the specific formula used to obtain the representative tokens, and the low-level offloading implementation together with cache" and it should be clearly framed as such. That being said, I do not think these are light contributions and once they are fully justified to be crucial to good performance, they yield a very impactful result.
> > >
> > > Also thank you for running the InfLLM on RULER but the numbers that reported are much lower than those reported in the original paper's Table 1. For example Mistral 7B obtains the average 68.4. What is the base model in this case?
> > >
> > > Regarding the offloading: Thank you for the explanation and reporting the effect of the cache. I think this will be a useful addition  to include in the paper. However, what I was asking about is the overhead of offloading itself. To be more precise, the question is what is the time difference between keeping everything on GPU and offloading to CPU (either with or without cache). I understand that as the length grows, keeping everything on GPU becomes impossible but I still want to understand what is the overhead of moving things to CPU.
> > >
> > > [3] Aminabadi RY, Rajbhandari S, Awan AA, Li C, Li D, Zheng E, Ruwase O, Smith S, Zhang M, Rasley J, He Y. Deepspeed-inference: enabling efficient inference of transformer models at unprecedented scale. InSC22: International Conference for High Performance Computing, Networking, Storage and Analysis 2022 Nov 13 (pp. 1-15). IEEE.
> > >
> > > [4] Zhang Z, Sheng Y, Zhou T, Chen T, Zheng L, Cai R, Song Z, Tian Y, Ré C, Barrett C, Wang Z. H2o: Heavy-hitter oracle for efficient generative inference of large language models. Advances in Neural Information Processing Systems. 2024 Feb 13;36.

---

> ### Author Response · Authors · 2024-08-12
>
> Thank you for your comments. I would like to re-emphasize the following points:
>
> 1. **Training-Free Context Extrapolation**: You mentioned that training-free is "a slightly different setting." I cannot agree with this view. As we discussed in the global response, training-free context extrapolation is a challenging and crucial task requiring further efforts and is parallel to training-based context extension. As large models continue to grow, the ability to extrapolate the context window size without training has become a crucial area of focus, receiving significant attention [1-6]. The most relevant works to our paper are those that apply sliding window attention in a training-free manner, such as LM-Infinite (NAACL 2024 Outstanding Paper) and StreamingLLM (ICLR 2024), both of which have garnered considerable attention and follow-up. All contributions in our paper are aimed at advancing the issue of training-free context extrapolation, not just improving the efficiency of LLMs in long text inference. We agree that memory-based models are widely applied in language models, but their application without training remains a significant, largely unexplored challenge.
>
> 2. **Importance of Training-Free Extrapolation**: As stated in our introduction and global response, training-free context extrapolation is a **crucial and challenging task**. We wish to further emphasize its importance:
>    - It holds potential for **real-world applications requiring unlimited streaming inputs**. As mentioned, any training-based model encounters challenges with out-of-distribution scenarios caused by unseen context lengths. Even a 128K LLM struggles in these streaming contexts.
>    - Training on long texts can adversely affect model performance on shorter texts [7]. Many studies have shown that training on long texts impacts short-text performance, a problem avoided in training-free settings.
>    - Continuing to pre-train on long texts is not the optimal approach. Creating diverse, high-quality long-text alignment datasets remains extremely difficult. For instance, even after extended pre-training of the Llama3-base model on long texts, we still lack the high-quality alignment data needed to achieve results comparable to Llama3-instruct.
>
>    To summarize, training-free context extrapolation is an important method for most researchers to get a well-performing long-text model. At the same time, all the above points cannot be solved by training-based methods.
>
> 3. **Formula of Our Approach**: Given the importance and challenges of training-free context extrapolation, suggesting a change in the formulas of InfLLM may not be a good choice. Our contributions are specifically within the context of training-free extrapolation, a key research direction parallel to training-based methods.
>
> 4. **Model Performance**: We compared InfLLM with full-attention models in the original paper and global response. Full-attention model can be regarded as the upper bound of long-text models. Our experiments on InfiniteBench have shown that our approach can achieve comparable performance with full-attention methods, even without training. Thus, we argue that our comparative experiments are sufficiently robust, demonstrating that our method approaches the effectiveness of full-attention models in a training-free setup. Therefore, compared to training-based memory method, InfLLM can achieve at least comparable performance with no additional training and can be applied in a plug-and-play manner for all transformer-based models, demonstrating its practicability.
>
> 5. **RULER Experiment**: The RULER experiment was conducted using Llama3 and focused on multi-step reasoning tasks in long texts, which explains the relatively lower accuracy.
>
> 6. **Offloading Experiment**: We have supplemented our paper with additional experiments on offloading. Here we present the relationship between the size of GPU cache and inference time costs of Passkey Retrieval. From the results, we can observe that the offloading mechanism will not incur much additional time costs with limited cache size.
>    | Size of GPU Cache | All   | 32    | 16    | 0     |
>    | ----------------- | ----- | ----- | ----- | ----- |
>    | Time (s)          | 28.80 | 28.89 | 29.33 | 33.50 |
>
> I appreciate your attention to these points and look forward to your reconsideration based on these clarifications. We will add the discussion in the revision.
>
> [1] Efficient streaming language models with attention sinks. ICLR 2024.
> [2] LM-Infinite: Zero-Shot Extreme Length Generalization for Large Language Models. NAACL 2024 (Outstanding Paper).
> [3] LLM maybe LongLM: Self-extend llm context window without tuning. ICML 2024.
> [4] Training-Free Long-Context Scaling of Large Language Models. ICML 2024.
> [5] Scaling laws of rope-based extrapolation. ICLR 2024.
> [6] A human-inspired reading agent with gist memory of very long contexts. ICML 2024.
> [7] LongRoPE: Extending LLM Context Window Beyond 2 Million Tokens. ICML 2024.

---

> > ### Comment · Reviewer_erR9 · 2024-08-14
> >
> > Dear Authors,
> >
> > I agree that extrapolation is definitely an important challenge. But there is no need to achieve extrapolation without training as long as you do not need to train at the target context length. There are several earlier work that require a bit of fine-tuning but then allow the model to extrapolate. So in these cases, you also can train at a context length such as 1024, and then do inference at 128k. One example is Memorizing Transformers [1]. I think not needing the fine-tuning is a nice bonus but unlike how you are trying to frame it, it is absolutely not essential to the problem of extrapolation. Again, since you keep emphasizing this point, I understand that training at the target inference length is not considered extrapolation and that is not what I am talking about here.
> >
> > > Formula of Our Approach: Given the importance and challenges of training-free context extrapolation, suggesting a change in the formulas of InfLLM may not be a good choice. Our contributions are specifically within the context of training-free extrapolation, a key research direction parallel to training-based methods.
> >
> > I am baffled by this response. Why is that not a good choice to suggest showcasing the importance of the chosen formula and what does it have to do with the importance of extrapolation?
> >
> > > We compared InfLLM with full-attention models in the original paper and global response. Full-attention model can be regarded as the upper bound of long-text models. Our experiments on InfiniteBench have shown that our approach can achieve comparable performance with full-attention methods, even without training.
> >
> > Yes but it is not clear whether the set of tasks you have chosen is suitable for correctly demonstrating long-context performance. This is an outstanding challenge to find good benchmarks for long-context. In the meantime, providing comparison with other long context methods can show how hard the task actually is and also provide the reader with a way to evaluate what amount of loss in precision is acceptable. For example, if all methods are able to get 99%, losing 0.1% accuracy is significant. Whereas if most methods only get 20% and full attention gets 90%, even a 70% accuracy is considered impressive.
> >
> > > The RULER experiment was conducted using Llama3 and focused on multi-step reasoning tasks in long texts, which explains the relatively lower accuracy.
> >
> > Can you elaborate on how using LLama3 explains the lower accuracy? the RULER paper reports a >80% average for LLaMA3.1 (8B) so I am not sure if the claim that the problem is the model not being able to do multi-step reasoning, is reasonable.
> >
> > > Offloading experiments
> > Thank you. This is indeed interesting and useful.
> >
> > I am maintaining my score given the above discussion.

---

> > > ### Author Response · Authors · 2024-08-14
> > >
> > > Thank you for your comments and critique regarding our manuscript. We appreciate your perspectives and acknowledge your concerns about the necessity of a training-free setting. The training-free paradigm is one that has garnered significant attention within the research community. Our decision to adopt this setting was informed by its emerging relevance and several unique benefits that we have detailed in our paper and responses. We believe this setting holds considerable potential for future developments in our field. We respect and value your opinion and we plan to expand our discussion on this topic in the revised manuscript. Moreover, we are eager to further explore these issues with you in person at the upcoming conference, as we believe they present valuable and promising avenues for research.

---

### Author Rebuttal · Authors · 2024-08-06

### Q1: Training-Free Context Extrapolation is Crucial
- Existing works for LLMs mainly focus on extending the model's context window through continued pre-training. However, these approaches face the following challenges:
   a) Long-sequence training requires substantial computational resources and large-scale high-quality datasets.
   b) The process of long-sequence training often leads to performance degradation on short-text data.
- In real-world applications, we envision LLMs capable of continuously processing unlimited streaming text input, such as in life-long personal assistants, LLM-driven agents. However, the training length of models will always be finite. Therefore, training-free context length extrapolation is crucial. Our method can handle streaming input, utilizing cost-effective CPU memory to store large key-value caches, allowing for scaling to longer texts.

### Q2: Training-Free Context Extrapolation is Challenging
- Effective block retrieval: As demonstrated in Focused Attention and Landmark Attention, differentiable block representation is critical for memory-based methods (Appendix C in Focused Attention, Section 4.2 in Landmark Attention). Our paper proposes a training-free block representation method that effectively mitigates the impact of irrelevant tokens on memory lookup.
- Efficient retrieval: Given the vast number of tokens in long contexts, token-level retrieval is prohibitively slow. Many existing training-based memory methods focus on constructing token-level memory units, such as unlimiformer and knn-LM, which usually require spending amounts of time for building retrieval index for large-scale tokens in each input sequence. It will lead to unaffordable time costs.
In summary, our paper focuses on the practical scenario of training-free context window extrapolation, proposing a memory module that requires no training and can effectively enhance the model's performance when processing streaming inputs. In the revision, we will add more discussion to clarify the contribution of our paper.

### Q3: Compared to Existing Memory-based Methods
As discussed in Q1 and Q2, training-free context extrapolation is crucial and challenging.
- Existing memory-based methods are mainly proposed for **pre-training language models from scratch**, which requires amounts of computation resources and human efforts.
- Besides, most existing memory-based methods focus on **token-level memory units** (KNN-Transformer, Unlimiformer, etc), which require a lot of time to build retrieval indexes for large-scale tokens in each input long sequence. We evaluate the time costs for token-level memory construction. For an LLM with token-level memories, it takes 2 hours to process a 128K sequence. Therefore, the cost of time is unaffordable.
- Some methods also adopt block-level memory (Landmark Attention, Focused Attention), these methods highlight the process of training effective block representations with long sequence data. From the evaluation on the Passkey retrieval task (Figure 1 in Focused Attention, Figure 3b in Landmark Attention), Landmark Attention can only achieve 80% accuracy for 32K sequences and Focused Attention can only achieve accuracy lower than 80% for 256K sequences.

### Q4: Compared to Well-Trained Long LLMs
In this study, we compared InfLLM with the continuously pre-trained Llama-3-8B-Instruct-Gradient-1048K. Furthermore, as suggested by Reviewer 5j15, we conducted a comparison combining InfLLM with Yi-9B-200K. The results are presented as follows. From the results, we can observe that InfLLM can also achieve comparable results with Yi-9B-200K.

|            | R.PK  | R.Num | R.KV | Choice | QA   | Sum  | Math.F | Avg. |
| ---------- | ----- | ----- | ---- | ------ | ---- | ---- | ------ | ---- |
| Yi-9B-200K | 100.0 | 98.3  | 54.5 | 63.3   | 13.0 | 5.9  | 23.4   | 51.2 |
| InfLLM     | 100.0 | 98.3  | 47.8 | 45.4   | 8.2  | 4.7  | 33.1   | 48.2 |

### Q5: Evaluation on RULER
LongBench and InfiniteBench are two widely used long-text evaluation datasets, encompassing tasks such as context retrieval, question answering, and summarization. To further demonstrate the effectiveness of InfLLM across various tasks, we conducted additional evaluations using the RULER benchmark with 128K tokens. The results are shown in the following Table. From the results, we can observe that InfLLM can still outperform the sliding window attention mechanism.

|        | NIAH-Single | NIAH-MQ | NIAH-MV | NIAH-MK | FreqWord | Var-Track | QA   | Commondword | Avg. |
| ------ | ----------- | ------- | ------- | ------- | -------- | --------- | ---- | ----------- | ---- |
| Stream | 5.3         | 4.9     | 5.3     | 3.5     | 90.1     | 10.0      | 20.8 | 0.1         | 17.5 |
| InfLLM | 43.2        | 8.3     | 9.0     | 5.3     | 89.8     | 15.1      | 19.6 | 0.1         | 23.8 |

---

### Decision · Program_Chairs · 2024-09-25

**Decision:**

Accept (poster)

**Comment:**

This paper proposes a new training-free context length extrapolation method for LLMs.

Intially, this paper had three marginal acceptance and one rejection scores.

By the rebuttal, one reviwer raised his/her score to 7, while other reviewrs kept their score.

For clear decision, AC carefully focused on the discussion between the authors and R-erR9.

It seems that both viewpoints are valuable and reasonable. In practical, in the viewpoint of LLM practices in real-world, fine-tuning-based context explortation apporaches are practical without requiring too heavy GPU resources.
However, academic research needs to explore different approach beyond practice for more fundamental solutions to make a breakthrough.

Considering the contributions of this paper and this perspective, this paper seems to satisfying the quality for NeurIPS.

So, AC recommends accepting this paper.
AC asks the augthors to address R-erRP's viewpoints properly in the cam-ready version.